Single exponential decay waveform; a synergistic combination of electroporation and electrolysis (E2) for tissue ablation

Klein Nina n.klein@biophysik.org 1 2
Guenther Enric 1 2
Mikus Paul 1
Stehling Michael K. 1 2
Rubinsky Boris 1 3
1 Inter Science GmbH , Gisikon , Switzerland
2 Prostata Center, Institut fur Bildgebende Diagnostik , Offenbach , Germany
3 Department of Mechanical Engineering, University of California , Berkeley , CA , United States
Zhang Yu-Dong
Electronic publication date: 2017 Apr 18
Publication date: 2017
Volume: 5
Electronic Location ID: e3190
Received 2016 Oct 10; Accepted 2017 Mar 15
Copyright: ©2017 Klein et al.
Copyright year: 2017
Copyright holder: Klein et al.
License: This is an open access article distributed under the terms of the Creative Commons Attribution License, which permits unrestricted use, distribution, reproduction and adaptation in any medium and for any purpose provided that it is properly attributed. For attribution, the original author(s), title, publication source (PeerJ) and either DOI or URL of the article must be cited.
License URL: https://creativecommons.org/licenses/by/4.0/

Keywords: Tissue ablation, Synergy electroporation and electrolysis, Liver, Electrolytic ablation, Reversible electroporation, Irreversible electroporation, Electrolysis

Funding: The authors received no funding for this work.

==============================
Background

Electrolytic ablation and electroporation based ablation are minimally invasive, non-thermal surgical technologies that employ electrical currents and electric fields to ablate undesirable cells in a volume of tissue. In this study, we explore the attributes of a new tissue ablation technology that simultaneously delivers a synergistic combination of electroporation and electrolysis (E2).

Method

A new device that delivers a controlled dose of electroporation field and electrolysis currents in the form of a single exponential decay waveform (EDW) was applied to the pig liver, and the effect of various parameters on the extent of tissue ablation was examined with histology.

Results

Histological analysis shows that E2 delivered as EDW can produce tissue ablation in volumes of clinical significance, using electrical and temporal parameters which, if used in electroporation or electrolysis separately, cannot ablate the tissue.

Discussion

The E2 combination has advantages over the three basic technologies of non-thermal ablation: electrolytic ablation, electrochemical ablation (reversible electroporation with injection of drugs) and irreversible electroporation. E2 ablates clinically relevant volumes of tissue in a shorter period of time than electrolysis and electroporation, without the need to inject drugs as in reversible electroporation or use paralyzing anesthesia as in irreversible electroporation.

Introduction

A number of biophysical and biochemical phenomena occur simultaneously when electric fields are applied across biological matter. These include Joule heating due to electrical current energy dissipation, electrolytic reactions at the interface between the electrodes and the biological milieu, and cell membrane permeabilization known as electroporation. All these electrical phenomena are used for tissue ablation. Usually the electrical potential delivery protocol is designed in such a way as to maximize one phenomenon, while minimizing the others. For example, in non-thermal irreversible electroporation (NTIRE) the electrical potential profile is designed to maximize irreversible electroporation while minimizing Joule heating (Davalos, Mir & Rubinsky, 2005). The non-thermal aspect of NTIRE was found to be beneficial in tissue ablation treatments, in which it is desired to spare vital sites in the treated lesion, such as blood vessels and nerves.

In electrolytic tissue ablation, cell death is caused by the chemical interaction between the products of electrolysis and cells (Nilsson et al., 2000; Czymek et al., 2011). Because the ablation is caused by a chemical reaction, it is a function of compounds concentration and time of exposure. One drawback of tissue ablation by electrolysis is the need for high concentrations of electrolytes and lengthy times of exposure. An advantage is the very low currents and voltages used.

In ablation by electroporation, brief, pulsed, high electric fields are used to permeabilize the cell membrane. Lower electric fields and small numbers of pulses yield reversible electroporation, in which the cell membrane permeabilization is temporary. Higher electric fields with larger number of pulses yield irreversible electroporation in which the cell membrane permeabilization is permanent, which results in cell death. Both reversible and irreversible electroporation are used for tissue ablation, each with their advantages and disadvantages. Reversible tissue electroporation is used for tissue ablation in combination with cytotoxic additives, in a procedure known as electrochemotherapy (Mir et al., 1991; Marty et al., 2006). One advantage of ablation by means of irreversible electroporation over electrochemotherapy is that no chemotoxic drugs are injected into the tissue (Rubinsky, Onik & Mikus, 2007), while the advantage of electrochemotherapy over irreversible electroporation is the use of fewer pulses and lower electric fields. The need to inject cytotoxic additives adds a complicating step to the electrochemotherapy procedure. Cell death through electrochemotherapy is dependent on mitosis cycle rendering and is possibly more tissue selective (Mir, Banoun & Paoletti, 1988; Orlowski & Paoletti, 1988), while irreversible electroporation induces apoptosis and necrosis instantaneously over the whole volume exposed to sufficiently high fields. However, the high electric fields and the large number of pulses used in conventional irreversible electroporation protocols cause some undesirable effects. They induce muscle contractions that require the use of a muscle relaxant and deep anesthesia during surgery. Every clinical electroporation protocols, reversible or irreversible, generates some products of electrolysis, and some heat (Turjanski et al., 2011; Maglietti et al., 2013). We have recently shown that if substantial amounts of products of electrolysis are inadvertently generated during an electroporation protocol, a highly detrimental electrical discharge across the layer of gas formed on the electrodes can occur (Guenther et al., 2015).

In several recent papers we have shown that combining electroporation and electrolysis (E2) sagaciously yields a new technology of tissue ablation with certain advantages over tissue ablation by electroporation (reversible or irreversible) or electrolysis alone (Phillips et al., 2015; Phillips et al., 2016; Stehling et al., 2016). We have developed several possible synergistic electroporation and electrolysis (E2) protocols. One effective combination entails delivering first several (eight) reversible electroporation type pulses followed by the injection of a low voltage direct current to generate products of electrolysis. While effective, this combination requires two different power supplies, one for electroporation and the second for electrolysis (Phillips et al., 2015; Phillips et al., 2016; Stehling et al., 2016). The combined voltage profile of electroporation pulses followed by low voltage electrolysis reminded us of an exponential decay waveform (EDW), generated by the discharge of a capacitor; a type of pulse which was rather common in the early stages of electroporation research (Sale & Hamilton, 1968). The shape of the capacitor discharge exponential decay waveform is a high initial voltage followed by a rapid decay towards a trailing low voltage. This type of waveform is still used in cell electroporation. We thought that with a properly chosen set of capacitor discharge parameters, the initial high voltage over a suitable timeframe could serve for electroporation, while the trailing lower voltage could generate sufficient charge for the generation of electrolytic products. The feasibility of tissue ablation with a EDW, was shown in the liver of a small rodent (Phillips et al., 2016).

The goal of this study was to extend the observations made in a small animal model and in an acute study (Phillips et al., 2016) to a larger animal model and a chronic study in order to show that EDW has the ability to ablate tissue volumes of clinical significance. The experimental study was supported by a first order mathematical analysis to evaluate the electric fields and extent of thermal damage generated by the exponential decay waveform.

Materials and methods

Animal protocol

The study was approved by Sir José Antonio Rodríguez Correa, Director of Animal Health Programs and General Director of Department of Agriculture and Livestock, Ministry of Environment and Rural, Agricultural Policies and Territory, Government of Autonomous Community of Extremadura (Spain), with application form number: 2015209030009567 and study register number: 100370001499. The experiment was conducted on in vivo pig liver, which was in accordance with Royal Decree Law 53/2013 (Feb.1st). According to the study protocol, three female pigs between 90 and 110 kg were treated. After being fasted for 24 h, animals were pre-medicated with a combination of diazepam (0.4 mg/kg) and ketamine (15 mg/kg) injected intramuscularly (IM). Anesthesia was induced with intravenous (IV) Propofol (3 mg/kg). Endotracheal intubation was performed and anesthesia was maintained with sevoflurane in oxygen (adjusted to 1.8–2% End tidal sevoflurane). Possible postoperative pain was treated with Buprenorphine 0.01 mg/kg IM Pre-med at recovery and Carprofen 4 mg/kg at extubation/recovery. Cefazolin 25 mg/kg IV was administrated every 2 h. If found to be needed during the procedure, the study had the ability to deliver pancuronium (0.1 mg/kg, at a dose of 1 mg/ml) through an IV to reduce muscle contractions during the application of the electrical pulses. The liver was exposed via a midline incision. The treatment was delivered using two 18-gauge Titanium needles (Inter Science GmbH, Ch) with a variable length (1–4 cm exposed treatment length) insulating sheath inserted in the liver. Titanium was chosen because, unlike steel or aluminum, it is chemically inert, is biocompatible and at room temperature inert to oxygen, chloride and corrosion (Emsley, 2001). The 18-gauge (1.02 mm) variable length electrodes were custom designed for the delivery of both electroporation and electrolytic pulse sequences.

The experiment was carried out in an open-surgery setting to maximize the availability of liver compartments. The delivery of the single exponential decay waveform solely took place through the described needle-type electrodes. This type of intervention in which only probes are inserted in tissue for ablation is defined as minimally invasive, as opposed to invasive resection surgery. This definition applies to cryosurgery, radiofrequency ablation and the various electroporation modalities.

Two electrodes were inserted in the liver under ultrasound monitoring, in a roughly axial parallel configuration, normal to the liver surface. Ultrasound images were also taken throughout the procedure. Since no apparatus is currently available to produce the exponential decay voltage waveform needed for the SEE procedure conceived by us, we have designed and built a new power supply described in the following device section. The parameters varied in this study were: the initial voltage and the time constants of the exponential voltage waveform. In addition, we varied the number of exponential voltage waveforms delivered. A total of 23 lesions were produced, in three pigs, in separate experiments. Animals were sacrificed at 24 h. The pigs were euthanized using Euthasol 2.2 ml/kg IV.

To fix the liver for microscopic viewing, a Foley catheter was placed into the descending aorta and the hepatic vein was snipped off for drainage of the affluent. The liver was flushed with physiological saline for ten minutes at a hydrostatic pressure of 80 mmHg from a pressurized IV drip. Immediately following saline perfusion, a 10% formalin fixative was perfused in the same way for ten minutes. The liver lobe in which the SEE lesion was made was removed and stored in the same formalin solution. For microscopic analysis, the tissue was bread loafed perpendicular to the capsule surface and parallel to the needle tracts. The needle tracts were marked on the liver surface and the exact location of each experiment was noted to be able to find the correct needle tracts. All cassettes were processed routinely from 10% phosphate buffered formalin to wax blocks. Five micrometer sections were made from each block and stained with Masson’s trichromatic stain for histologic examination. The stained samples were examined and analyzed by an independent histology service company and reports were prepared (Narayan Raju, Inc., South San Francisco, CA, USA). The focus of the histology was to verify the extent and nature of tissue ablation with E2. To produce information of practical clinical value, the focus of the analysis was on verifying the ability to produce a continuous lesion between the electrodes.

Device

We were unable to find a power supply that can produce the waveform parameters, required for an EDW protocol in tissues with the dimensions of the pig liver. Therefore, we designed a power supply that operates in the modality of capacitor discharge electroporation systems (e.g., Gene Pulser Xcell™ Electroporation System; BioRad, Hercules, CA, USA) with an enhanced performance. The conventional type capacitors used were replaced with a 100 microfarad capacitor to provide the charge required for electrolysis. Similar to the Gene Pulser Xcell™, the generator has an output of up to 3 kV. Because of the larger capacitors it can generate exponential decay waveforms up to time ranges of hundred milliseconds, depending on tissue conductivity and thereby simultaneously deliver electrolysis and electroporation. The apparatus selects and matches the internal components needed to produce the time constants selected for the specific tissue conductivity of the treatment area by selecting an appropriate capacitor. The apparatus is able to produce and deliver the exponential decay voltage profile in the time and voltage range for the specific treatment area. Figure 1B illustrates typical exponential decays shapes obtained for the in Table 1 listed electrical parameters of resistance and capacitance. Resistance and capacitance fully define the electrical components of the device.

Figure 1 (A) Generator and data acquisition schematic. (B) An illustrative waveform applied in pig liver for 1,000 V and a time constant of 37 ms including statistical details of the measurement and fit from the DAC. (C) Needle configuration.

Table 1 Parameters used for the thermal and electrical field calculations.

Initial voltage U0	750 V (Fig. 3), 1,000 V (Figs. 4 and 5), 1,500 V (Fig. 6)	
Exposure length	1 cm (Figs. 3–5) 2 cm (Fig. 6)	
Decay	Exponential capacitor discharge	
Power supply capacitance	Listed in figures	
Distance between electrodes	1.5 cm	
Electrode diameter	1 mm	
Liver: electrical conductivity	0.286 S/m	
Liver: heat capacity	3,750 J/(kg*K)	
Liver: density	1,000 kg/m3	
Liver: thermal conductivity	0.52 W/(m*K)	
Titanium: electrical conductivity	7.4e5 S/m	
Titanium: heat capacity	710 J/(kg*K)	
Titanium: density	4,940 kg/m3	
Titanium: thermal conductivity k	7.5 W/(m*K)	

Mathematical analysis

The conductivity of the tissue in the model was adjusted to best fit the measured discharge curve of the capacitors of each pulse, which, since the charge of the capacity was defined, also defines the current. An illustrative waveform is shown in Fig. 1. While much more advanced mathematical models for electroporation and their thermal effects during pulse delivery are available (Corovic et al., 2013) we used a simplified model for several reasons. First, the available models do not have an exponential decay form yet. Second, electrolysis may substantially change the electrical and thermal parameters of the tissue, in a way that is not yet understood. Also, there are not described effects on the metallic surface of the electrode changing conductivity. Therefore, we have used a simplified first order model which can provide the necessary information needed for this study. For future studies, the conductivity increase due to Joule heating and electroporation should be taken into account. The thermal and electrical field simulations in this study were performed using a finite element solver (Comsol Multiphysics 5.2) for the Laplace equation (electrical field) and Pennes Bioheat equation, in a way identical to that described in (Davalos, Mir & Rubinsky, 2005) in a 3D model. The setup was approximated as two parallel titanium cylinders in a large volume of liver tissue with the parameters shown in Table 1. In case of discharging capacitors, the amount of Joules heating in tissue is prescribed by the dissipation of the charge energy, Q, (Q = C∗U0). Therefore, specifying only, the initial voltage (U0) and capacity (C) is sufficient to simulate the experiment. Thermal damage begins at temperatures higher than 42 °C, but only for prolonged exposures on the order of several seconds to hours. Damage is relatively low until 50–60 °C at which the rate of damage dramatically increases (Diller, 1992). Our interest was not in the distinction between these two processes, but only in the “worst case scenario” to estimate a radius of damage that could have been caused by thermal effects. We chose 30 s to look at a “worst case scenario” of tissue damage by heat to observe a lager distance from the electrode due to heat dissipation.

A total of 30 s appeared to be the worst case in terms of radius of possible thermal induced tissue necrosis when using the perfusion parameters as cited. The waveform delivered to the electrodes was assumed to be a perfect exponential decay in time, t, (U = U0∗exp − (t∕τ)), where U0 is the initial voltage and the time constant is, τ. The time constant was taken from the experimental data, through the analysis of the voltage trace during the delivery of the waveform.

Results

A series of 23 lesions were generated in experiments in which we studied the effects of the E2 waveform parameters on tissue ablation. The study examined the effects of the initial voltage, the time constant and the number of exponential decay voltage waveforms delivered. To facilitate a systematic and well defined analysis of the E2 phenomenon, we will focus on the results at midline between the two electrodes. The parameters chosen for this study were drawn from the experimental results of (Phillips et al., 2016) and are listed in Table 2. They were chosen to deliver substantial amounts of electrolytic products.

Table 2 Relevant parameters of the displayed experiments.

Lesion #	U/V	E-field/V/cm	Distance/mm	Exposure/mm	C/µF	τ/ms	Comment	
1	750	500	15	10	100	50	Fig. 2C	
2	750	500	15	10	100	100	Fig. 2D	
3	1,000	667	15	10	100	70	Fig. 3	
4	2 × 1,000	2 × 667	15	10	100	79, 92	2 pulses 30 s interval, same polarity, Figs. 4 and 5	
6	1,500	1000	15	20	100	70	Figs. 6 and 7	

Figure 2 Study with an EDW with initial voltage difference between electrodes of 750 V and various time constants.

(A) Calculated electric field. (B) Calculated thermal field after 30 s. (C) Macroscopic image—50 ms time constant—no ablation was noticed. (D) Macroscopic image—100 ms time constant—some ablation near electrodes.

Figure 2 shows results from a series of studies in which the initial voltage between electrodes was 750 V, the distance between electrodes was 15 mm, the exposed length was 10 mm and the depth of penetration was 20 mm. This configuration produces an initial voltage over distance of 500 V/cm. The calculated electrical field norm is displayed in Fig. 2A and the calculated temperature in Fig. 2B. Geometrically, both graphs represent the 1d-cutline through the perpendicularly induced electrodes with the electrical field and the temperatures respectively on the y-axis. Figures 2C and 2D are the macroscopic histology from lesions treated with a voltage of 750 V between the electrodes and time constants of 50 ms and 100 ms, respectively. If tissue resistance and conduction between electrode and tissue were constant, the discharge could be fully described using the time constant of the EDW. However, secondary effects like thin layers of burned tissue, can cause insulation and hence disrupt the ideal exponential decay. This does not necessarily have any negative effect on the ablation, but will limit τ to adequately describe the delivered waveform. The panels show the formalin embedded samples, sectioned in a plane that is transverse to the centers of the two electrodes. In all the different experiments with 750 V (500 V/cm voltage to distance between electrodes) there was no configuration in which the lesion between electrodes became continuous. Figures 2A and 2B show that at the line midway between the electrodes the electric field is less than 200 V/cm and the temperature is below 40 °C.

Figure 3 shows results from an experiment in which the initial voltage between electrodes was 1,000 V, the distance between electrodes was 15 mm, the exposed length was 10 mm, the depth of penetration was 20 mm and the time constant was 70 ms. The slides were prepared with Masson’s trichrome staining. Figure 3A gives an overview of the evaluated slide. The image is taken in a plane that transverses the centers of the two electrodes. The area of the probe is clearly visible, with a deep blue color at the site of the probes, representing the cellular damage caused by thermal necrosis, surrounded by areas of coagulated blood (deep red color). The 10× magnification at the anode (Fig. 3B) illustrates an area of thermal necrosis, where the hepatocytes have sustained more intense cellular ablation injury resulting in denaturation of the cytoplasmic organelles. At the cathode (Fig. 3D) we can witness the gradual effect of the treatment: Around the macroscopically visible lesion there is a pale area which represents less affected cells immediately adjacent to the severely affected hepatocytes (marked with an arrow). The sinusoidal spaces are dilated due to edema and/or hepatocellular swelling, while the nuclei are condensed. The space between the electrodes is not fully ablated, as the microscopic images show areas of unaffected cells (Fig. 3C). Figure 3E shows the calculated electric field for a voltage of 1,000 V and Fig. 3F shows the calculated temperature distribution. Figures 3E and 3F show that, for these experimental conditions, the minimal electric field midway between the electrodes is calculated to be about 240 V/cm and the temperature midway between the electrodes is well below 40 °C.

Figure 3 Study with one EDW with initial voltage difference between electrodes of 1,000 V and time constant of 70 ms.

(A) Histological slide with Masson’s trichrome staining. (B) 10× magnification of the right lesion, which is the anode. We see severe acute hepatocellular necrosis with coagulated blood (hemorrhage) in the sinusoids. (C) 10× magnification between the electrodes. The cells do not appear to be affected. (D) 10× magnification at the margin of the left lesion, which is the cathode. Here we see the borderline between the necrotic tissue on the left and partially affected cells on the right. (E) Electric field strength distribution. (F) Temperature distribution after 30 s (scale bar 100 µm).

Figure 4 illustrates the histology of liver, from a treatment in which two voltage exponential decay waveforms with similar parameters as those that produced Fig. 3, were delivered at an interval of 30 s. The macroscopic image taken from a plane between the center of the two electrodes (Fig. 4A) shows that the partial electrode pathway (tunnel) is filled with coagulated blood. This is confirmed by the deep red linear region in the histological slides stained with Masson’s trichrome staining in Figs. 4B–4D. The dark blue zone around that region (Figs. 4B–4D) represents the more severely ablated hepatocytes, by virtue of being closest to the point of energy release. Figure 4E shows the calculated electric field for an exponential decay waveform with an initial voltage of 1,000 V and Fig. 4F shows the calculated temperature distribution at the onset of the second pulse. There are two aspects to notice in Figs. 4E and 4F. Figure 4E is a copy of Fig. 4E. It is obvious because we have used the electrical parameters of normal liver. However, it is known that the electrical conductivity of electroporated tissue changes after electroporation (Sel et al., 2005; Ivorra & Rubinsky, 2007), and therefore this panel may not be correct. We also anticipate that electrolysis will affect the electrical and thermal parameters in a way that is not fully known. The second aspect relates to the temperature distribution. Figure 4E shows that the calculated temperature distribution, when the second pulse is delivered is substantially elevated over the initial temperature when the first pulse is delivered, and thermal damage may be induced near the electrodes.

Figure 4 Study with 2 EDW separated by 30 s with time constants of 79 and 92 ms, the first and second pulse respectively, 1,000 V difference between electrodes placed at a distance of 15 mm between them, 10 mm exposed length, 100 µF capacitor. Liver was extracted 18.5 h after treatment.

(A) Macroscopic histological slide (cathode left electrode, anode right electrode). (B) Masson’s trichrome staining reveals blood coagulation (red) and ablation both around and in between electrodes. (C) Close-up of the cathode, which is the left electrode. (D) Close-up of the right electrode, which is the anode (scale bar 500 µm). (E) Electric field strength distribution. (F) Temperature distribution prior to the delivery of the second waveform at 1 and 30 s.

Figure 5 shows 10× magnified images of the histological slide from Fig. 4. Figure 5A shows the space between the electrodes. Fig. 4B gives a 10× magnification of that area, showing a full ablation zone, with affected cells throughout the area. Hepatocytes both at the cathode (Fig. 5C) and anode (Fig. 5D) show condensed nuclei, with hemorrhage in the spaces between, however with intact vessels (Fig. 5C).

Figure 6 shows the histological results of exponential voltage profile in which the initial voltage between electrodes was 1,500 V, the distance between electrodes was 15 mm, the exposed length was 20 mm and the depth of penetration was 30 mm. It is important to notice that the shown top 10 mm of the electrode was insulated. The slides were prepared with Masson’s trichrome staining. Figure 6A shows the cells on the center line between the electrodes at the level of the top 10 mm insulated part of the electrodes. Here we see that the cells are not affected by the treatment. Figure 6B, however, shows the lesion which was caused by the treatment in the uninsulated part of the tissue between the electrodes. The lesion is continuous between electrodes at this level. Figure 6C displays the calculated electric field for a voltage of 1,500 V, and Fig. 6D shows the calculated temperature distribution. The electric field midway between the electrodes is about 550 V/cm. The midway between electrodes temperature is about 40 °C and way above 50°C in proximity of the electrodes.

Figure 5 Details from Fig. 4.

(A) Space between the electrodes in Fig. 4. Bar indicates 500 µm. (B) 10× magnification of cells between the electrodes, showing the details of the ablated area. (C) 10× magnification of the cathode, showing edema and cellular ablation injury. (D) 10× magnification of the area by the anode, showing the margin of affected and non-affected cells. All images show Masson’s trichrome staining. Bars in B–D indicate 100 µm.

Figure 6 Study with a EDW with a time constant of 69 ms, 1,500 V difference between electrodes placed at a distance of 15 mm between them, 200 mm exposed length, 100 µF capacitor.

(A) Macroscopic cross section in a plane through the axis of the electrodes. Image taken between electrodes at the part where the electrodes were insulated, showing that the cells are not affected. (B) Image taken between electrodes where the electrodes were not insulated showing that the lesion was bridged (500 µm bar). (C) Electric field strength distribution. (D) Temperature distribution after 30 s.

Figure 7 displays a 10× magnification of the pathological slide shown in Fig. 6. Figure 7A is a magnification of the cathode, showing swollen and necrotic hepatocytes and a disrupted sinusoidal pattern. Between the electrodes (Fig. 7B) a bridged ablation with affected cells was observed, with a complete loss of cellular structure. At the anode (Fig. 7C) there is an affected cellular architecture with hemorrhage. Figures 7A–7C show open and undamaged large blood vessels within the treatment field.

Figure 7 10× magnification of the pathological slides shown in Fig. 6.

(A) Image taken at the right electrode, which was the cathode, showing necrotic, swollen hepatocytes and a disrupted sinusoidal pattern. (B) Image taken between the electrodes, illustrating a complete loss of cellular structure with swollen hepatocytes. (C) Left electrode, which was the anode, showing an affected cellular architecture and hemorrhage. Note that the large blood vessels are open and unaffected. Scale bar 100 µm.

Discussion

Our main criteria for evaluating the exponential decay voltage waveform ability to ablate tissue in a clinically significant manner was the ability to induce the ablation throughout the gap between the electrodes. Therefore, the histological and mathematical analysis is focused on the tissue found midway between the electrodes. This is the part of the treated tissue in which the lowest electric fields and lowest temperatures occur. Figures 2 and 3 show that there are parameters of initial voltage and time constant for which the tissue midway between the electrodes is not ablated. Figure 2A shows that for an initial voltage of 750 V and a distance of 1.5 cm between the electrodes (500 V/cm distance between electrodes), the electric field strength midway between the electrodes is lower than 200 V/cm. This value is substantially below the reversible electroporation threshold for the rabbit liver, which was measured to be 362 +/21 V/cm (Miklavčič et al., 2000). Since the calculated temperature midway between electrodes is below 40 °C, there is no mechanism to induce damage between the electrodes. The conditions in the region between the electrodes are below the levels required for irreversible electroporation ablation, reversible electroporation or thermal ablation.

Our experiment was designed to be performed without a muscle relaxant, however, in such a way as to allow for an immediate use of a muscle relaxant as soon as an undesirable level of muscle contraction is noted. From among the 23 experiments with the exponential decay voltage waveform done in this pig liver study, a muscle contraction requiring the use of a muscle relaxant (pancuronium) was detected in none. Two of our researchers (MS and PM) have experience with several hundred animal and human IRE procedures. On a scale from 1 to 5, for IRE muscle contraction without muscle relaxants, they evaluate the contractions we observed as less than one. In comparison with muscle contractions when a muscle relaxant is used, they evaluated the contractions as similar. In fact, the muscle contraction was negligible, and at times unnoticeable. It should be emphasized that the reduction in muscle contraction is an expected benefit from the use of a single pulse. Reducing muscle contractions in IRE is an important area of current research. In particular, the HFIRE technology developed by Davalos and his group (Siddiqui et al., 2016) and nanosecond pulses technology (Schoenbach et al., 2007). We believe that the effect we observed is related to the electrical discharge across electrolytically produced layer of gas around the electrodes. We have shown in (Guenther et al., 2015) that the electrical discharge across this layer of gas plays a major contribution to the observed violent motion of the electroporated object. We have also shown that this motion occurs primarily during the later pulses in a series of pulse experiments, when the electrolytically produced gas layer becomes substantial. Our single exponential decay pulse, while generating product of electrolysis, eliminates the violent electrical discharge across that gas layer. The violent discharge is eliminated, because by the time a large layer of gas has formed near the electrode, the potential at the electrode is below the value that can induce electrical breakdown. Obviously this observation is relevant only to the parameters used in this study, in which the maximal voltage was 1,500 V (1,000 V/cm voltage over distance) and the maximal time constant 148 ms.

Figure 3 shows that increasing the initial voltage of the exponential decay waveform to 1,000 V will also increase the extent of the damage near the electrodes. The distance between the electrodes is 1.5 cm and therefore the initial voltage to distance ratio is 750 V/cm. Figure 3E shows that the electric field midway between electrodes is calculated to be below 300 V/cm. This value is below the 362 ± 21 V/cm reversible electroporation threshold (Miklavčič et al., 2000). Tissue damage by heat can be also excluded, since the temperature between electrodes does not exceed 40 °C (Figs. 2B and 3F). In this case, the conditions in the middle between the electrodes are also below the levels required for irreversible electroporation ablation, reversible electroporation or thermal ablation.

Figures 4 and 5 show that it is possible to ablate the entire zone between electrodes by using two consecutively delivered exponential decay waveforms with the same electrical parameters, distance between the electrodes and waveform shape as those used to produce the results in Fig. 3. The initial voltage of the exponential decay waveform was 1,000 V, and as the distance between the electrodes is 1.5 cm, the initial voltage to distance ratio is 750 V/cm. Figure 3 shows that cells between the two electrodes survive the delivery of a single exponential decay waveform. The difference is that the results in Fig. 3 were obtained with one exponential decay waveform and those in Figs. 4 and 5 were obtained with two exponential decay waveforms. This suggests that the delivery of the second waveform is responsible for the cell death in the zone between the electrodes. Figures 3E and 4E show the theoretical calculated electric field. It is seen that in the zone between the electrodes the electric field is typical of reversible electroporation. This may explain why cells survive in the middle zone in Fig. 3. There are a few possible explanations for the mechanism of cell death in the central zone between the electrodes after the delivery of two exponential decay waveforms. One explanation may be the change in electrical and biophysical properties as a consequence of the delivery of the first waveform and the attendant change in the electrical field that was actually delivered during the second waveform. Figure 4F shows that the calculated temperature prior to the delivery of the second exponential waveform is elevated relative to that prior to the delivery of the first waveform. Elevated temperatures favor electroporation and may reduce its threshold (Polak et al., 2014). Furthermore, it is known that electroporation changes the electrical conductivity of tissue. While Fig. 4E was obtained for the electrical conductivity of the normal liver, the second waveform may generate a somewhat modified electric fields, which may favor the cell death seen in the middle zone in Figs. 4 and 5. An additional or another possible mechanism responsible for the difference in cell ablation between the cases depicted in Figs. 3 and 4 may be related to the effect of electrolytic products. The second waveform, which is responsible for the cell ablation in the central zone in Fig. 4, has delivered twice the level of electrolytic compounds than that delivered in the single waveform treatment whose results are depicted in Fig. 3. It is possible that the enhanced cell death may be related to the increased amount of products of electrolysis, due to the second waveform and their synergistic effect with reversible electroporation. A combination of the effects of changes in properties and products of electrolysis may be also responsible for the difference in extent of cell ablation between the results depicted in Figs. 2 and 3. Obviously a more thorough study is needed to elucidate this mechanism. An important aspect of this study would be a detailed mathematical model that combines calculation of electric fields, temperature and effects of electrolysis.

The results displayed in Figs. 6 and 7 produce stronger evidence of the E2 mechanism of tissue ablation. Here, an increase of the exponential decay waveform initial voltage to 1,500 V has produce ablated tissue between the electrodes. Calculations show that the electric field midway between the electrodes is about 550 V/cm (Fig. 6E). This value is below the irreversible electroporation threshold for the rat liver (637 V/cm ± 43 V/cm)(Miklavčič et al., 2000), keeping in mind that this threshold depends on several parameters such as pulse shape, number of pulses, pulse duration and configuration (Qin et al., 2013). The temperature midway between electrodes is about 40 °C, (Fig. 6F), which is below the threshold of thermal damage. The mechanism of tissue ablation at the midpoint between electrodes is neither irreversible electroporation nor thermal. The most likely possible mechanism is the synergistic effect of electrolysis and reversible electroporation.

The E2 protocol requires a special waveform comprised of an exponential decay shape with a steep decrease in voltage to values that will not induce an electrical discharge across the electrolytically product near the electrodes and a longer low voltage tail, that can generate sufficient products of electrolysis for the E2 ablation. To the best of our knowledge currently available electroporation systems cannot deliver exponential decay waveforms with the desired, electrolytic products generating time constants. To this end we have modified existing commercial designs (e.g., Gene Pulser Xcell™ Electroporation System; BioRad, Hercules, CA, USA), as described in the methods and materials section. The key difference is the use of larger capacitance, in essentially the same circuit.

This is a first large animal study on the use of the synergy between electrolysis and reversible electroporation to enhance tissue ablation by electroporation. However, the E2 combination seems promising. It has the ability to create comparable clinically relevant areas of tissue ablation, in a much shorter period of time than irreversible electroporation, with lower voltages and single waveforms, without the need to inject drugs and without the need for paralyzing anesthesia.

We would like to thank Dr Narayan Raju from Pathology Research Laboratory, Inc. for his assistance on the pathological examination and analysis.

Additional Information and Declarations

Competing Interests

Author Contributions

Animal Ethics

Data Availability

The authors are employees of InterScience GmbH, Luzern, Switzerland—a company in the field of tissue ablation by electroporation. Boris Rubinsky is an Academic Editor for PeerJ.

Nina Klein and Enric Guenther performed the experiments, analyzed the data, wrote the paper, prepared figures and/or tables, reviewed drafts of the paper.

Paul Mikus and Michael K. Stehling performed the experiments, reviewed drafts of the paper.

Boris Rubinsky conceived and designed the experiments, performed the experiments, analyzed the data, wrote the paper, reviewed drafts of the paper.

The following information was supplied relating to ethical approvals (i.e., approving body and any reference numbers):

The study was approved by Sir José Antonio Rodríguez Correa, Director of Animal Health Programs and General Director of Department of Agriculture and Livestock, Ministry of Environment and Rural, Agricultural Policies and Territory, Government of Autonomous Community of Extremadura (Spain), with application form number: 2015209030009567 and study register number: 100370001499.

The following information was supplied regarding data availability:

The raw data is included in Table 2 in the manuscript.

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
