# Peer review of "Single exponential decay waveform; a synergistic combination of electroporation and electrolysis (E2) for tissue ablation"

_PeerJ, doi:10.7717/peerj.3190_

## Round 0.1 · original submission · Major Revisions

· Academic Editor

Major Revisions

Please highlight your changes in the revised version.

Reviewer 1 ·

Basic reporting

Authors revised the manuscript very carefully.

Experimental design

Experimental work has not well defined in the manuscript.

Validity of the findings

Authors did not mention the validation of work in detail.

Additional comments

Manuscript ID: #13740
Review Letter
The paper presents single exponential decay waveform; a synergistic combination of electroporation and electrolysis for tissue ablation. The work is focused on the experimental work on the pig liver for treatment during thermal ablation. Some shortcomings of the paper are lack of interesting to reader.

The referee’s comments are given as follows:

Single exponential decay wave is not well defined in the manuscript. There is no any support of the behalf of electrolytic ablation and electroporation based ablation are minimally invasive. Authors have some lines ‘A new device that delivers a controlled dose of electroporation field and electrolysis currents in the form of a single exponential decay waveform…’ but work has no any sporting argument.

These are main drawbacks of the manuscript so that this is not acceptable in present form.

Annotated reviews are not available for download in order to protect the identity of reviewers who chose to remain anonymous.

·

Basic reporting

The mansucript is fairly well written, but could be organised better. i have made some commenst and suggestions where and how in my opinion the manuscript could be modified for better clarity. I am missing trace/recordings (at least as an example) of the voltage and current applied. The motivation for the study should be given more clearly.

Experimental design

The experimental design is not always clear. Authors should make effort to explain why different time constants were used, how and why 30 s was selected, why 40 and 50 degress Celcius is used as "treshold" - is it becouse these are round numbers or becouse of protein denaturation, cell death or some other reason: it is also not clear why some experiments were performed with single and some with two pulses.

The modeling, i.e mathematical analysis of electric field is not up-to-date.

Validity of the findings

Validity of findings regarding heart and muscle contraction and use of muscle relaxants appear as important concluison, but i am not convinced that experimeantal design and results allow this kind of conclusion to be made.

Additional comments

see my comments and suggestions in the attached file

Reviewer 3 ·

Basic reporting

Clear, unambiguous and professional English used throughout the manuscript.
Figures are relevant, of proper quality and well labelled and described. However, some captions (e.g. “Electric field norm” of fig 1) maybe too small if the figures are scaled down in the final publication. Some arrows (or other signs) on the histological slides would also help to clarify the location of the electrodes and the observed effects on the cells.
One of the advantages of the proposed method (E2) over NTIRE would be the avoidance of neuromuscular stimulation and, therefore, the need for muscle relaxants. However, this is an issue already addressed by Prof. Davalos et al. with novel pulse protocols for IRE (H-FIRE) and by other researchers using nanosecond pulses. In my opinion, these other research efforts should be cited and compared with E2 in terms of advantages and disadvantages.
Please add citations to support “Cell death through electrochemotherapy is dependent on mitosis cycle rendering and is possibly more tissue selective,” (lines 74-75)
Please add citations to support “(Titanium) is chemically inert, and does not introduce toxic metals in tissue, during the electrolysis stage. ,” (lines 135-136)
Not all raw data seems to be supplied. This is a requirement of PeerJ. For instance, a few histological sections are displayed in the manuscript whereas 23 lesions were generated. Please clarify (or add raw data). In particular, please add the initial voltages and the time constants for each treatment (and the currents, if measured)
Line 81: protocols -> protocol

Experimental design

Please better clarify whether a single pulse (exponential decay waveform) is applied per treatment or more than one are applied.
Please include many more details regarding the generator (e.g. switching mechanism, load dependence…) and the applied voltage waveforms (e.g. voltage and current magnitudes, waveforms…). A coarse schematic for the generator would be helpful. Illustrative waveforms would also be very helpful.
Where (institution) the experimental study was carried out?

Validity of the findings

The threshold cited for irreversible electroporation of the rat liver (637 V/cm +/- 43 V/cm)(Miklavcic et al. 2000) most likely corresponds to 100 microsecond pulses whereas in this case exponential decay waveforms are employed. Longer pulses are known to cause electroporation with lower electric fields. Therefore, it is not valid to use the above threshold (or any threshold obtained with short rectangular pulses) to rule out the occurrence of IRE midway between electrodes when exponential decay waveforms are employed. This compromises the validity of the findings: it cannot be concluded that the observed effects necessarily correspond to the synergistic combination of electroporation and electrolysis. These effects could correspond to electroporation alone. In my opinion this is a major issue in the present manuscript and study that requires amendment.

---

## Round 0.2 · Minor Revisions

· Academic Editor

Minor Revisions

Please revise your paper to address the comments from the two reviewers.

·

Basic reporting

Authors have responded to my comments and suggestions in satisfiyng manner.
The manusrcipt is clear, references are up to date and relevant.

Experimental design

Experiments and analysis of results is adequate. There ar two minor comments that i have identified:

Line 208: authors state »there is no point in using a model that is more complex than the first order«. I think this statement is to blunt and should be made milder. As it is know it conveys a message which is at lease debatable if not wrong. I cannot understand why the authors argue so strongly about this where it would be enough to state that their simple model provide necessary information – and admit that conductivity increase due to Joule heating and electroporation (at least) should be taken into account somewhere in future.

Line 445-453: but authors could also use high voltage followed by low voltage electric pulse – there are several devices on the market that deliver these kind of application. And parameters of pulses would be well defined – if not better.

Validity of the findings

Authors provide novel results in a sound way. Conclusions are clearly stated and are in accordance with original research question.

Additional comments

The authors have responded adequately to most of my questions and comments. There are some minor details that I would alert them to and i think it would be in the interest of readers to correct&ammend/respond to

Line 141: the diameter of electrodes is given as 18-gauge. Please provide (also) mm diameter.

Line 141: Authors state: »the electrodes made of Ti are MRI compatible«. Was MRI performed with these electrodes? I did not see results on this.

Lines 162 -172: I am missing description on how the exact location of needle/electrodes was determined in histology

Line 192: authors claim that the »amount of electrolytic species and also the Joule heating is predefined«. I am not sure I agree entirely with this statement. Namely the tissue conductivity is known to increase due to electroporation AND heating – so if the changes are not taken into account how can the amount of electrolytic species and also the Joule heating be predefined?

Line 253: delete »difference« - voltage is by definition the difference between potential in A and B

Line 299: at least for thermal parameter it is known how conductivity changes

Line 425: authors state »elevated temperature favor electroporation and may reduce its threshold« - can you please provide reference for this statement.

Line 429-431: i am sorry but i do not understand the logic behind this statement. Please consider rewriting.

·

Basic reporting

The English used in the text is clear and very professional.
The introduction lacks on some relevant cites regarding previous studies of pH and electroporation. For example; in line 76,77 “Every clinical electroporation protocols, 77 reversible or irreversible, generates some products of electrolysis, and some heat.”. Please add them.
Figures should state clearly the isoline drawn in blue in the electric field graph.
In line 183, says “Figure 1” and it should say “Figure 1 B”.
In line 270 says “pathology” and it should say “histology”

Experimental design

The research is original and within the scope of the journal. It shows a new technique for tissue ablation based on the combination of electroporation and electrolytic ablation with many significant advantages over other methods that should be explored further.
The detail in the materials and method section is adequate though I would suggest to add a table with pulse parameters on each experiment and the results in the results section. This will provide more clear data to reproduce this work.

Validity of the findings

The results are interesting and combine a new with an older technique improving both.
The data is clear and supports the results obtained.
Conclusions are adequate.

---

## Round 0.3 · accepted · Accept

· Academic Editor

Accept

Thanks for your submission

·

Basic reporting

the revised version of the manuscript is acceptable for publication. authors have adequately addressed all my questions.

Experimental design

the revised version of the manuscript is acceptable for publication. authors have adequately addressed all my questions.

Validity of the findings

the revised version of the manuscript is acceptable for publication. authors have adequately addressed all my questions.

Additional comments

the revised version of the manuscript is acceptable for publication. authors have adequately addressed all my questions.